# Gauge-Strain-Controlled Air and PWR Fatigue Life Data for 304 Stainless Steel—Some Effects of Surface Finish and Hold Time

**Marc Vankeerberghen [1],\*, Michel De Smet [2] and Christian Malekian [2]**

1   SCK CEN, Nuclear Materials Science Institute, 1170 Brussels, Belgium
2   ENGIE, Tractebel Engineering, 1000 Brussels, Belgium; michel.desmet@tractebel.engie.com (M.D.S.); christian.malekian@tractebel.engie.com (C.M.)
\*   Correspondence: Marc.Vankeerberghen@sckcen.be; Tel.: +32-14333182

**Abstract:** We performed environmental fatigue testing in simulated primary water reactor (PWR) primary water and reference fatigue testing in air in the framework of an international, collaborative project (INCEFA-PLUS), where the effects of mean strain and stress, hold time, strain amplitude and surface finish on fatigue life of austenitic stainless steels in light water reactor environments are being studied. Our fatigue lives obtained on machined specimens in air at 300 °C lie close to the NUREG/CR6909 mean air fatigue curve and are in line with INCEFA-PLUS air fatigue lives. Our environmental fatigue lives obtained in simulated PWR primary water at 300 °C lie relatively close to the NUREG/CR6909 mean fatigue curve; derived from the NUREG/CR6909 mean air fatigue curve and the applicable environmental correction factor (Fen). The PWR results show that (1) a polished surface finish has a slightly higher and a ground surface finish a slightly lower fatigue life than the NUREG/CR6909 prediction; (2) the ratio of polished to ground specimen life is ~1.37 at 300 °C and ~1.47 at 230 °C; (3) holds—at zero strain after a positive strain-rate—have a slightly detrimental effect on fatigue life. These results are in line with the INCEFA-PLUS PWR fatigue lives. A novel gauge-strain extensometer was deployed in order to perform a true gauge-strain-controlled fatigue test in simulated PWR primary water.

**Keywords:** environmental fatigue; 304 stainless steel; air; PWR primary water; 300 °C

## 1. Introduction

Environmentally-assisted fatigue (EAF) is worldwide a hot topic since the introduction of the requirement in USNRC Regulatory Guide 1.207 to address it [1]. Originally only for new reactors and long term operation LTO, it now also needs to be considered for 10-year license renewals. Current regulation is based on a state-of-the-art review of light water reactor fatigue data and effects [2].

Sparked by EPRI roadmap and gap analysis reports on EAF [3–5] and industry needs, a consortium of 16 organizations from across Europe participated in the five-year EAF project INCEFA-PLUS [6–10]. The effects of mean strain and stress, hold time, strain amplitude and surface finish on fatigue life of austenitic stainless steels in light water reactor environments have been studied; these being issues of common interest to all participants and mentioned as gaps to be addressed in [3–5].

In a previous paper [11], our first few fatigue tests under simulated PWR conditions were reported and their outcome and quality discussed.

In this paper, we report on further fatigue tests, performed in air and in simulated primary water of a pressurized water reactor (PWR), and on our analysis of (1) our fatigue data and (2) of fatigue data downloaded from the INCEFA-PLUS database as per September 2019. More extensive data analysis of the latter fatigue data is ongoing and being reported in [12,13] and references therein.

## 2. Materials and Methods

### 2.1. Material

The common material within the framework of environmental fatigue testing for INCEFA-PLUS was AISI 304 stainless steel. Details on the material can be found in [11]. All specimens that are labelled SC- further on, were made of this material.

For some air fatigue tests, an in-house 304 material was used. The material was prepared in the following way. First, parallelepipeds were machined out of 304 stainless steel material. These parallelepipeds were then heat-treated for half an hour at 1100 °C and water quenched. All specimens that are labelled 14xx further on, were made of this material.

The chemical compositions of both are given in Table 1, together with 304 and 304L specifications. One element, sulfur, was considered for affecting environmental fatigue life in [2], but only for carbon and low-alloy steel and only through the environmental correction factor $F_{en}$. Hence, we assume its influence on the air fatigue results here to be negligible. Other elements are not considered in [2] for affecting either air or environmental fatigue life within the stainless steel material group. Hence, we assume their influence on the air fatigue results here to be negligible.

**Table 1.** Chemical composition of the AISI 304 material (wt.%).

| Heat | Specimen | C | Cr | Cu | Mn | Mo | N | Ni | P | S | Si |
|---|---|---|---|---|---|---|---|---|---|---|---|
| INCEFA-PLUS | SC- | 0.029 | 18 | 0.02 | 1.86 | 0.04 | 0.056 | 10 | 0.029 | 0.004 | 0.37 |
| SCK CEN | 14xx | 0.031 | 18 | 0.36 | 1.56 | 0.27 | 0.082 | 8 | 0.040 | 0.021 | 0.36 |
| Specification 304L – | | <0.03 | 18-20 | | <2 | | <0.1 | 8–12 | <0.045 | <0.03 | <0.75 |
| Specification 304 – | | <0.08 | 18-20 | | <2 | | <0.1 | 8–10.5 | <0.045 | <0.03 | <0.75 |

### 2.2. Specimen Geometry

The design of the specimen for fatigue testing is given in Figure 1. Most, but not all, recommendations of ASTM—E606 were taken into account [14]. The gauge length to diameter ratio is 2.2 (within the ASTM range 3 ± 1). The transition region radius to diameter ratio is 4 (within the ASTM range 4 ± 2). The diameter is 4.5 mm and is smaller than the recommended minimum value of 6.35 mm because of load restriction on our machine for fatigue testing (±8 kN).

### 2.3. Surface Finish

Details on surface finish can be found in [11]. Polished INCEFA-PLUS specimens have a surface roughness with an Rt (maximum roughness height) of 0.762 µm and a corresponding Ra (arithmetic mean value) of 0.044 µm. Ground INCEFA-PLUS specimens have a surface roughness with an Rt of about 20 µm (14–50 µm) and a corresponding Ra of about 1.5 µm (1.1 µm to 5.1 µm). Table 2 shows measured roughness values for the ground specimens.

The in-house fatigue specimens were machined out of heat-treated parallelepipeds by lathe turning. One of them, specimen 1427, was measured, by a Handysurf E-35B instrument (ACCRETECH, Tokyo Seimitsu Co., JAPAN), to have a surface roughness with an Ra (arithmetic mean value) of 0.60 µm.

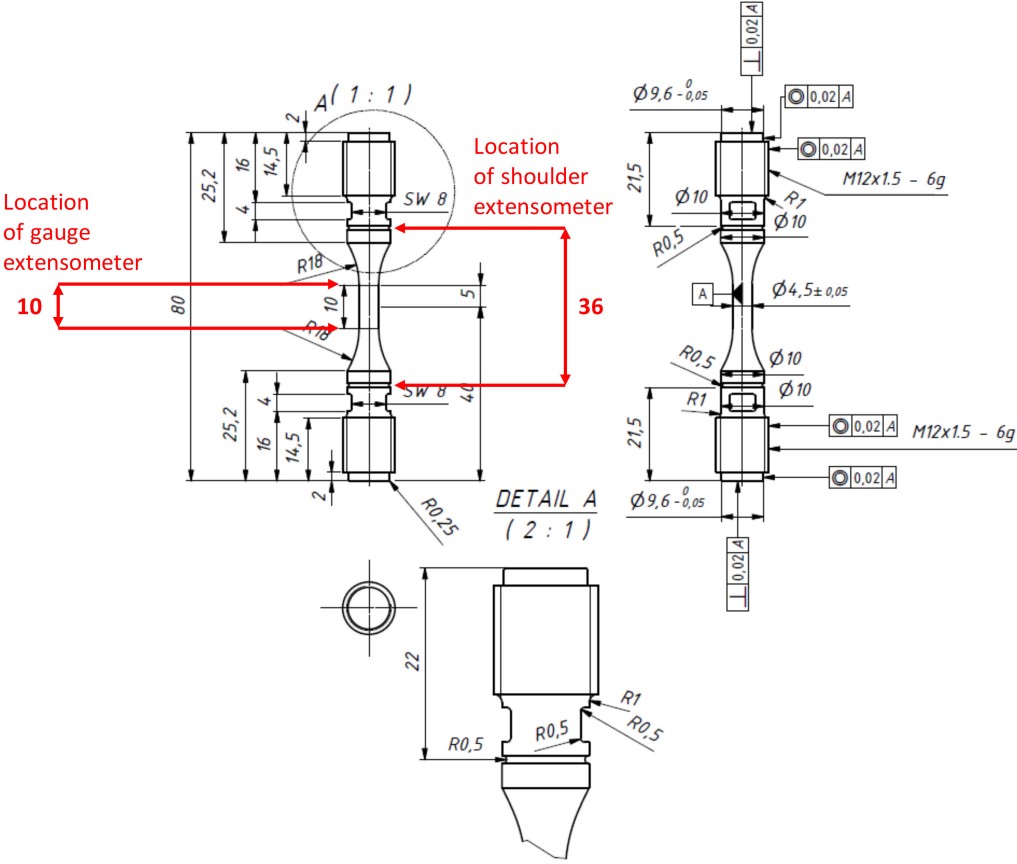

**Figure 1.** Specimen design and location of gauge and shoulder extensometers (dimensions in mm).

**Table 2.** Surface roughness (ground specimens).

| Specimen- | $R_t$ (µm) | $R_a$ (µm) | Specimen- | $R_t$ (µm) | $R_a$ (µm) | Specimen- | $R_t$ (µm) | $R_a$ (µm) |
|-----------|-----------|-----------|-----------|-----------|-----------|-----------|-----------|-----------|
| SC-3 | – | 1.1 | – | – | – | – | – | – |
| SC-5 | 13.7 | 1.5 | SC-17 | 49.1 | 5.1 | SC-30 | 25.0 | 3.2 |
| SC-6 | 14.6 | 1.4 | SC-18 | 40.4 | 3.5 | SC-33 | 22.7 | 3.2 |
| SC-8 | 15.2 | 1.5 | SC-19 | 39.0 | 4.1 | SC-34 | 28.3 | 3.6 |
| SC-9 | 15.9 | 1.6 | SC-31 | 27.4 | 3.3 | SC-35 | 23.9 | 3.2 |

## 2.4. Environment

The environment, to which the fatigue specimens were exposed, was either air at 300 °C or simulated primary water of a pressured water reactor (PWR). PWR primary water is high-temperature (300 °C), high-pressure (150 bar) water. It contains boron, added as boric acid (here at a concentration of 1000 ppm boron), lithium, added as lithium hydroxide (here at a concentration of 2 ppm lithium) and dissolved hydrogen (here at a concentration of 25 cc(STP)H2/kg). Further details on PWR water chemistry and the water chemistry loop can be found in [11].

## 2.5. Fatigue Testing

The environmental fatigue tests performed within the framework of the INCEFA-PLUS project were strain-controlled fatigue tests. They were executed following a procedure agreed between the various partners [15]. The procedure is based on international strain-controlled fatigue testing and related standards [14,16–19], to which environment specific details were added.

The waveform is saw-tooth or triangular. Nomenclature related to the waveform is shown in Figure 2; left for the controlling strain signal and right for the stress response signal. The positive strain rate is set to 0.01%/s. The negative strain rate is set to −0.1%/s or slower if required for proper PID control tuning. Specified strain amplitudes were 0.3% and 0.6%, well into the low-cycle fatigue regime. Some tests included holds (three holds of 72 h each). The three hold positions are at mean strain during a positive strain rate (Figure 3) and after a specified number of cycles has been reached (Table 3). The holds were spaced depending on the anticipated NUREG/6909 fatigue life (see Table 4). Holds were included into the inter-laboratory, INCEFA-PLUS test matrix to represent fatigue that is interrupted by holding for extensive periods of time at nominal plant conditions [7–10], an area for which there is a lack of data [3,5].

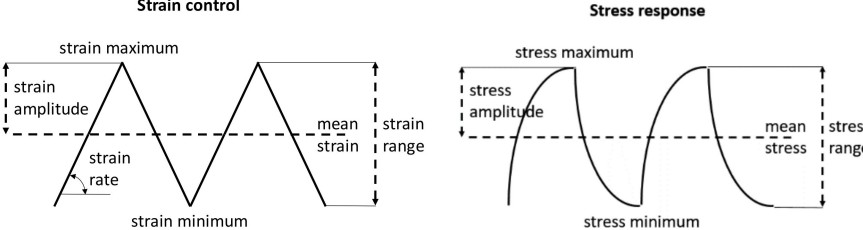

**Figure 2.** Nomenclature related to the waveform.

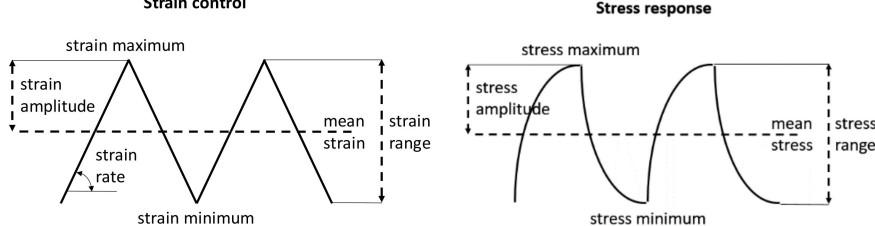

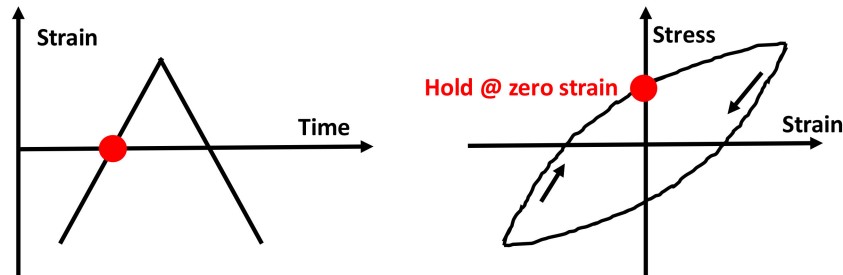

**Figure 3.** Location of INCEFA-PLUS holds within the hysteresis loop.

**Table 3.** Cycle number after which a hold takes place.

| Strain Amplitude | 0.3% | 0.6% |
|---|---|---|
| AIR | 6000, 12,000 and 18,000 | 1000, 2000 and 3000 |
| PWR | 1200, 2400 and 3600 | 200, 400 and 600 |

**Table 4.** NUREG/CR6909-estimated $N_{25}$ fatigue life.

| Strain Amplitude | Air | PWR 300 °C | PWR 230 °C |
|---|---|---|---|
| 0.3% | 24,592 | 5381 | 9142 |
| 0.6% | 3914 | 856 | 1455 |

*2.6. Extensometers*

Ideally, the strain must be measured, and controlled, over the gauge section of the specimen. In practice many partners testing on solid specimens in a PWR environment measure, and control, from shoulder to shoulder. We have installed two extensometer systems simultaneously; a shoulder extensometer and a gauge extensometer (for location, see Figure 1). Further details on the dual extensometer system can be found in [11]. Details on the gauge extensometer system can be found in [20].

### 3. Results

Our air and environmental fatigue test results are listed in Table 5, in terms of $N_{25}$ fatigue life. Fatigue life results already reported in [11] are indicated with a *. $N_{25}$ is determined based on a 25% drop in the cyclic maximum stress. Table 4 list expected $N_{25}$ fatigue life as per NUREG/CR6909 [2]. There, the mean fatigue life in air and up to 400 °C for austenitic stainless steel is given by Equation (1) and the mean fatigue life in a PWR environment is obtained by applying an environmental correction factor Fen, as illustrated in Equation (2). The environmental correction factor for 300 °C (230 °C) and a strain rate of 0.01% is 4.57 (2.69) for austenitic stainless steel in a PWR environment.

$$\ln\left(N_{25}^{air}\right) = 6.891 - 1.920 \ln(\varepsilon_a - 0.113) \tag{1}$$

$$N_{25}^{PWR} = \frac{N_{25}^{air}}{F_{en}} \tag{2}$$

**Table 5.** $N_{25}$ results for strain-controlled, air and environmental, fatigue tests.

| Specimen | Temperature | Surface Roughness | Mean Strain | Hold | Average Strain Amplitude (%) | Measured Fatigue Life |
|---|---|---|---|---|---|---|
| | | | Air | | | |
| SC-3 | 300 | ground | 0.5% | no hold | 0.622 | 3659 |
| 1414 | 300 | machined | no mean | no hold | 0.618 | 4140 |
| 1415 | 300 | machined | no mean | no hold | 0.379 | 11,288 |
| 1416 | 300 | machined | no mean | no hold | 0.303 | 28,616 |
| 1418 | 300 | machined | no mean | 3 × (72 h hold) | 0.600 | 4202 |
| 1420 | 300 | machined | no mean | 3 × (72 h hold) | 0.303 | 20,962 |
| 1421 | 300 | machined | no mean | 3 × (72 h hold) | 0.303 | 21,358 |
| | | | PWR | | | |
| SC-5* | 300 | ground | no mean | no hold | 0.312 | 4651 |
| SC-6* | 300 | ground | mean | no hold | 0.306 | 4387 |
| SC-8* | 300 | ground | mean | 3 × (72 h hold) | 0.296 | 3855 |
| SC-2* | 300 | polished | no mean | no hold | 0.309 | 6011 |

**Table 5.** *Cont.*

| Specimen | Temperature | Surface Roughness | Mean Strain | Hold | Average Strain Amplitude (%) | Measured Fatigue Life |
|---|---|---|---|---|---|---|
| SC-9* | 300 | ground | mean | 3 × (72 h hold) | 0.320 | 4219 |
| SC-13 | 300 | polished | no mean | no hold | 0.549 | 1388 |
| SC-17 | 300 | ground | no mean | no hold | 0.611 | 707 |
| SC-18 | 300 | ground | no mean | no hold | 0.302 | 4471 |
| SC-19 | 300 | ground | no mean | 3 × (72 h hold) | 0.622 | 649 |
| SC-15 | 300 | polished | no mean | no hold | 0.305 | 6262 |
| SC-16 | 300 | polished | no mean | no hold | 0.614 | 931 |
| SC-23 | 300 | polished | no mean | no hold | 0.614 | 868 |
| SC-31 | 300 | ground | no mean | 3 × (72 h hold) | 0.313 | 2980 |
| SC-24 | 300 | polished | no mean | no hold | 0.309 | 5748 |
| SC-25 | 300 | polished | no mean | 3 × (72 h hold) | 0.311 | 5296 |
| SC-26 | 300 | polished | no mean | 3 × (72 h hold) | 0.320 | 4989 |
| SC-41 | 300 | polished | no mean | 3 × (72 h hold) | 0.295 | 6564 |
| SC-30 | 230 | ground | no mean | no hold | 0.319 | 5000 |
| SC-33 | 230 | ground | no mean | no hold | 0.303 | 6307 |
| SC-34 | 230 | ground | no mean | no hold | 0.313 | 4481 |
| SC-28 | 230 | polished | no mean | no hold | 0.313 | 7990 |
| SC-29 | 230 | polished | no mean | no hold | 0.300 | 9607 |
| SC-39 | 230 | polished | no mean | no hold | 0.302 | 8576 |
| SC-40 | 230 | polished | no mean | no hold | 0.307 | 6746 |
| SC-35 | 230 | ground | no mean | no hold | 0.309 | 5357 |

## 4. Discussion

### 4.1. Air Fatigue Results at 300 °C

The obtained $N_{25}$ fatigue lives lie close to the NUREG/CR6909 mean air curve (Figure 4). This was not expected since the specimens had a machined surface finish, whilst the mean air curve is based on fatigue tests with polished samples. Polished samples have a lower surface roughness than machined samples (here Ra = 0.044 and 0.6 μm, respectively). Given that the machining marks were clearly visible and radial on the machined specimens and that the ground INCEFA-PLUS specimens show a reduced fatigue life (having a surface roughness between 1.1 and 5.1 μm Ra), one could have reasoned machined life to be closer to ground life than polished life on the basis of surface roughness.

### 4.2. Analysis of INCEFA-PLUS Air Fatigue Results

The INCEFA-PLUS air fatigue database (as at 17–09–2019) has been analyzed with respect to the effect of surface finish and holds on $N_{25}$ fatigue life. The analysis shows that:

(1) The air fatigue live of a polished *surface finish* is on average about 13% higher than that of a ground surface finish (Figure 5, 1.039/0.923) and this difference is found to be statistically significant. Prior to INCEFA-PLUS this number was expected to be larger. Since surface effects in NUREG/CR6909 Rev. 1 are in the range 1.5–3.5 (mean of 2.29) and that here it is evaluated at 1.13 the surface finish effect in NUREG/CR6909 Rev. 1 is somewhat overestimated in respect of INCEF-PLUS air fatigue data.

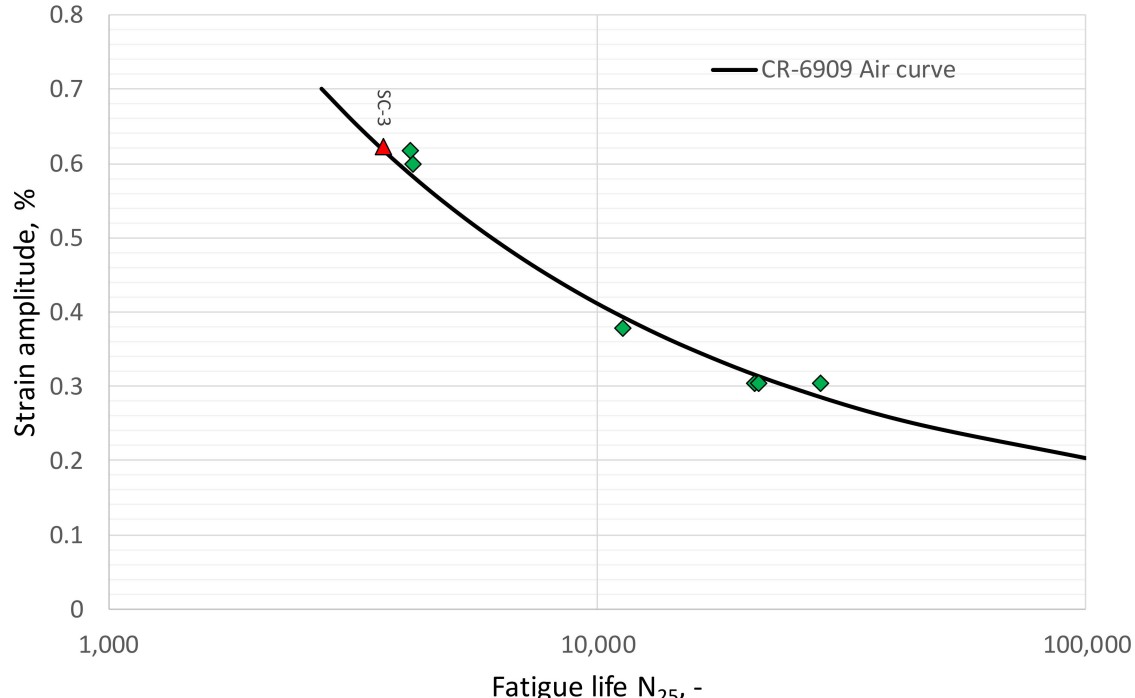

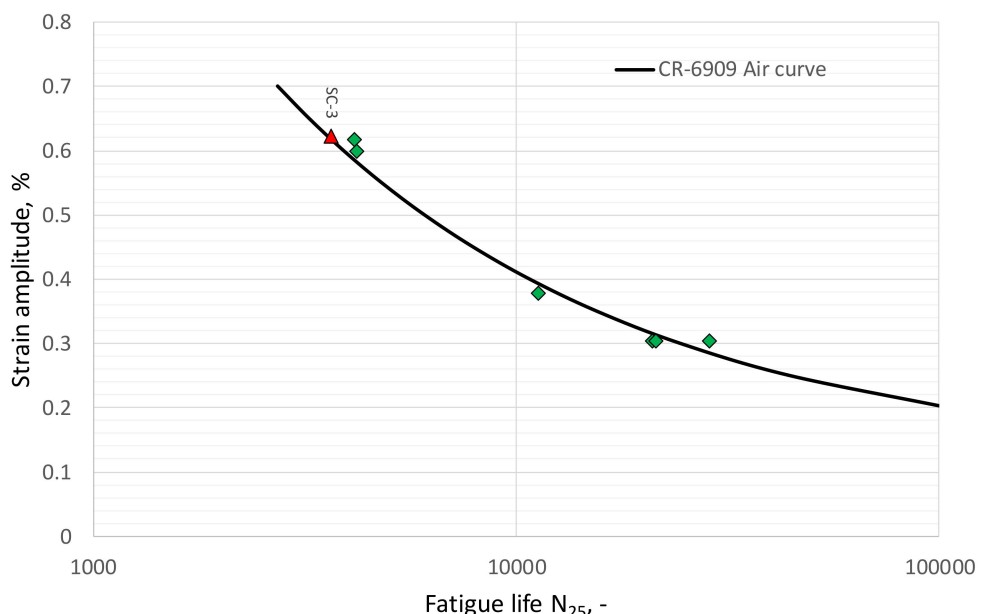

**Figure 4.** Air fatigue life results and NUREG/CR6909 mean air curve.

(2) The air fatigue live of a test with *holds* is on average about 5% lower than that of a test without holds (Figure 6, 0.945/0.996) but this difference is not found to be statistically significant. Prior to INCEFA-PLUS the effect was thought to be the opposite and larger. This indicates that a potential conservatism in the application of NUREG/CR6909 does probably not stem from holds during plant operation.

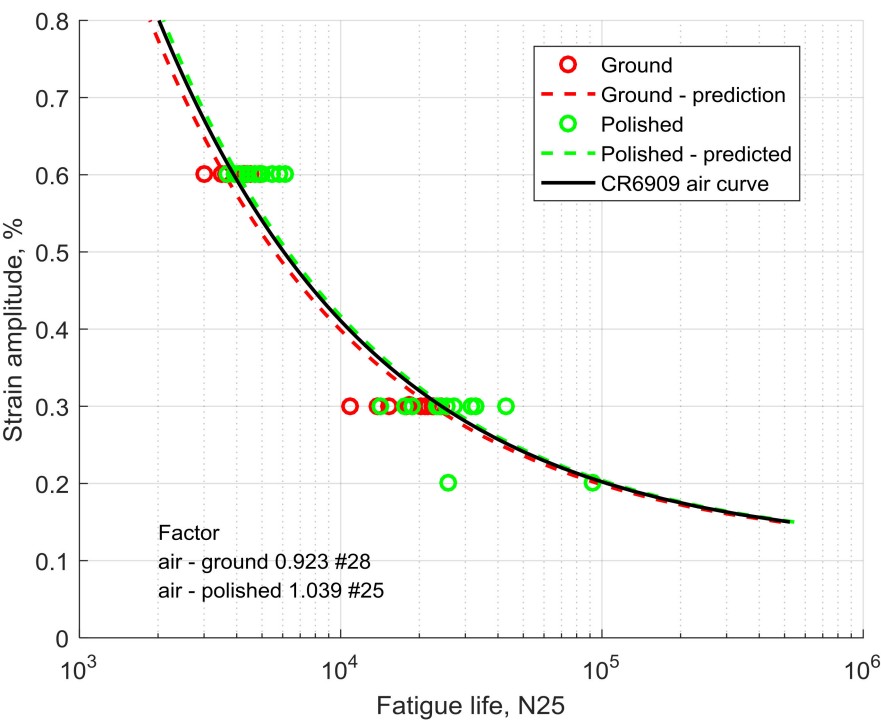

**Figure 5.** Effect of surface finish on INCEFA-PLUS air fatigue life results (data as per 17 September 2019).

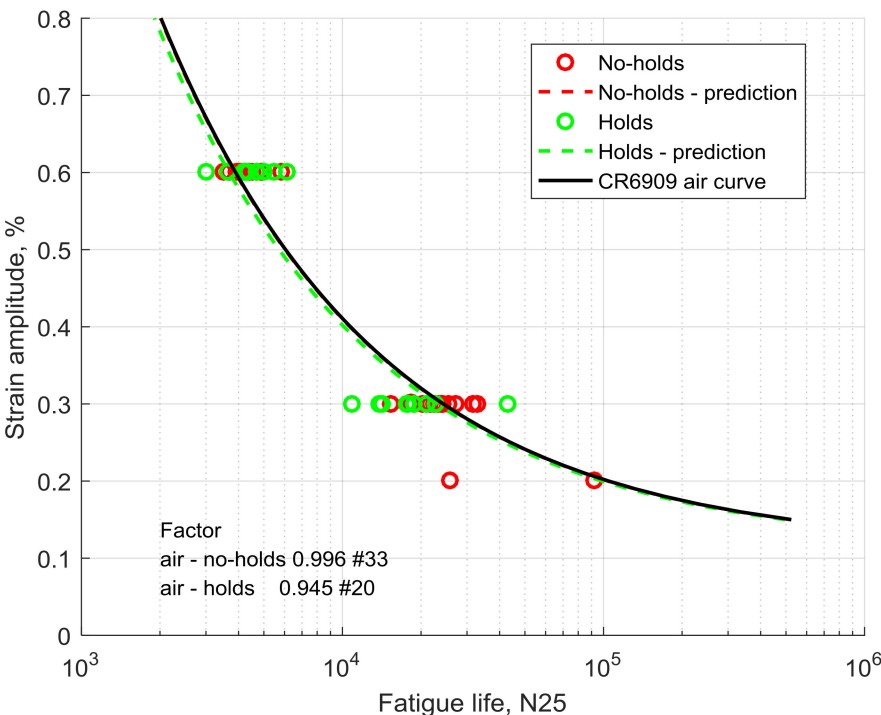

**Figure 6.** Effect of "INCEFA" holds on INCEFA-PLUS air fatigue life results (data as per 17 September 2019).

In Figures 5 and 6, the factor is a factor-on-life when compared to the mean CR6909 air curve. The # corresponds to the number of data points.

The statistical analysis is performed on a factor distribution, one for each test results. The factor is the ratio of the test life, $N_{25}$, to the CR6909 expected life, $N_{25}^{air}$ (see Equation (1)). It is assumed that the

natural logarithm of the factor is normally distributed. Further conclusions are that the polished, hold and no-hold data subsets are not statistically significant different from the CR6909 mean air curve but that the ground data subset is.

### 4.3. Effect of Surface Finish on PWR Fatigue Life at 300 °C

The green curve in Figure 7 that is labelled 'Fen-fit polished' is associated with the green circles and corresponds to a curve obtained from the CR6909 mean air curve (Equation (1)) with a best-fit factor of 3.87 on life. Hence, the PWR fatigue life for *polished* specimens is higher than the one predicted from NUREG/CR6909 [2], where the Fen factor would be 4.57 at 300 °C and 0.01%/s (black curve).

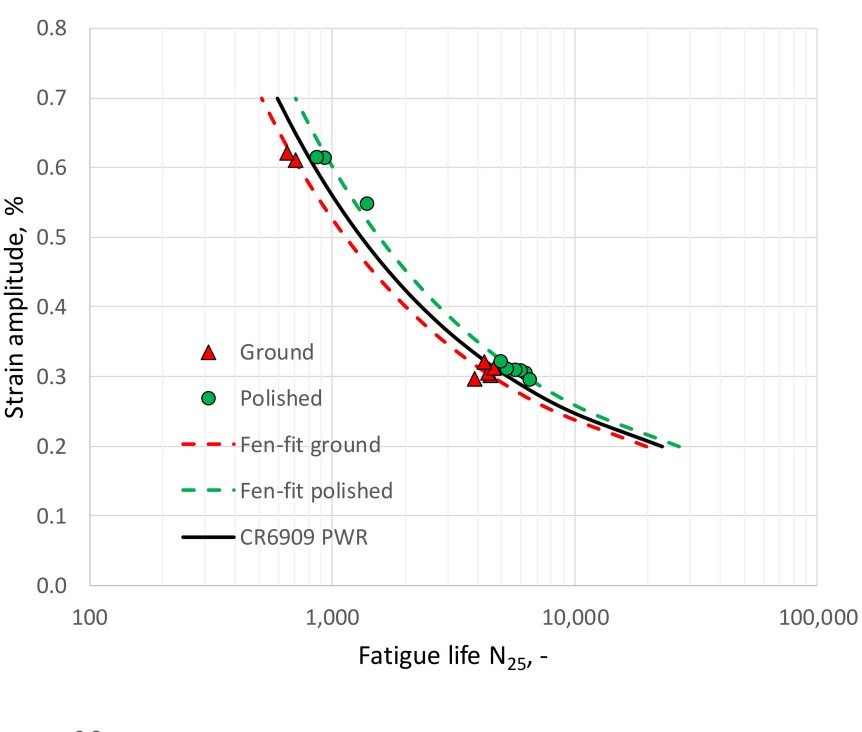

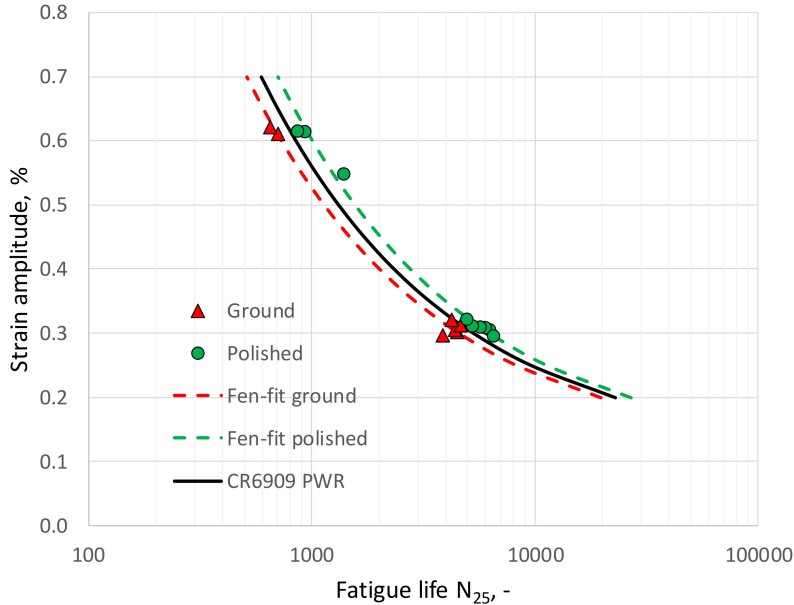

**Figure 7.** Primary water reactor (PWR) fatigue life results (by surface finish) and NUREG/CR6909 mean PWR curve at 300 °C (Fen = 4.57).

The red curve in Figure 7 that is labelled 'Fen-fit ground' is associated with the red triangles and corresponds to a curve obtained from the CR6909 mean air curve (Equation (1)) with a best-fit factor of 5.32 on life. Hence, the PWR fatigue life for *ground* specimens is lower than the one predicted from NUREG/CR6909 [2], where the Fen factor would be 4.57 at 300 °C and 0.01%/s (black curve).

The PWR fatigue life of a polished specimen is a factor 5.32/3.87 = 1.37 longer than the one of a rough specimen.

### 4.4. Effect of Holds on PWR Fatigue Life at 300 °C

The green curve in Figure 8 that is labelled 'Fen-fit no holds' is associated with the green circles and corresponds to a curve obtained from the CR6909 mean air curve with a best-fit factor of 4.29 on life. Hence, the PWR fatigue life for experiments without holds is slightly higher than the one predicted from NURGE/CR6909 [2], where the Fen factor would be 4.57 at 300 °C and 0.01%/s (black curve).

The red curve in Figure 8 that is labelled 'Fen-fit holds' is associated with the red triangles and corresponds to a curve obtained from the CR6909 mean air curve with a best-fit factor of 5.02 on life. Hence, the PWR fatigue life for experiments with holds is slightly lower than the one predicted from NURGE/CR6909 [2], where the Fen factor would be 4.57 at 300 °C and 0.01%/s (black curve).

The PWR fatigue life of experiments without holds is a factor 5.02/4.29 = 1.17 longer than the one of experiments with holds.

### 4.5. Effect of Surface Finish on PWR Fatigue Life at 230 °C

The green curve in Figure 9 that is labelled 'Fen-fit polished' is associated with the green circles and corresponds to a curve obtained from the CR6909 mean air (Equation (1)) curve with a best-fit factor of 2.80 on life. Hence, the PWR fatigue life for *polished* specimens is about the one predicted from NUREG/CR6909 [2], where the Fen factor would be 2.69 at 230 °C and 0.01%/s (black curve).

The red curve in Figure 9 that is labelled 'Fen-fit ground' is associated with the red triangles and corresponds to a curve obtained from the CR6909 mean air curve (Equation (1)) with a best-fit factor of 4.13 on life. Hence, the PWR fatigue life for *ground* specimens is lower than the one predicted from NUREG/CR6909 [2], where the Fen factor would be 2.69 at 230 °C and 0.01%/s (black curve).

The PWR fatigue life of a polished specimen is a factor 4.13/2.80 = 1.47 longer than the one of a rough specimen.

### 4.6. Our PWR Fatigue Results in the Light of Other INCEFA-PLUS PWR Fatigue Results

Within INCEFA-PLUS many more partners have been performing environmental and air fatigue testing [12,13]. We also performed analysis for the effect of surface finish on these test as per September 2019. Here we report on findings which, based on a simple t-test, yield significant effects. The statistical analysis is performed as describe at the end of Section 4.2.

Our surface finish analysis, on both our and INCEFA-PLUS obtained PWR fatigue lives, shows that:

(1) At 300 °C, the fatigue life of the polished specimens is slightly better than the fatigue life predicted by NUREG/CR6909. Indeed, the 90% confidence interval for the "CR6909" factors on life are 0.82–0.88 for our results and 0.83–0.99 for the INCEFA-PLUS results. Both ranges lie below 1, indicating some conservatism in NUREG/CR6909.

(2) At 230 °C, the fatigue life of the polished specimens corresponds to the fatigue life predicted by NUREG/CR6909. Indeed, the 90% confidence interval for the "CR6909" factors on life are 0.95–1.05 for our results. The range is around 1, indicating correspondence to NUREG/CR6909.

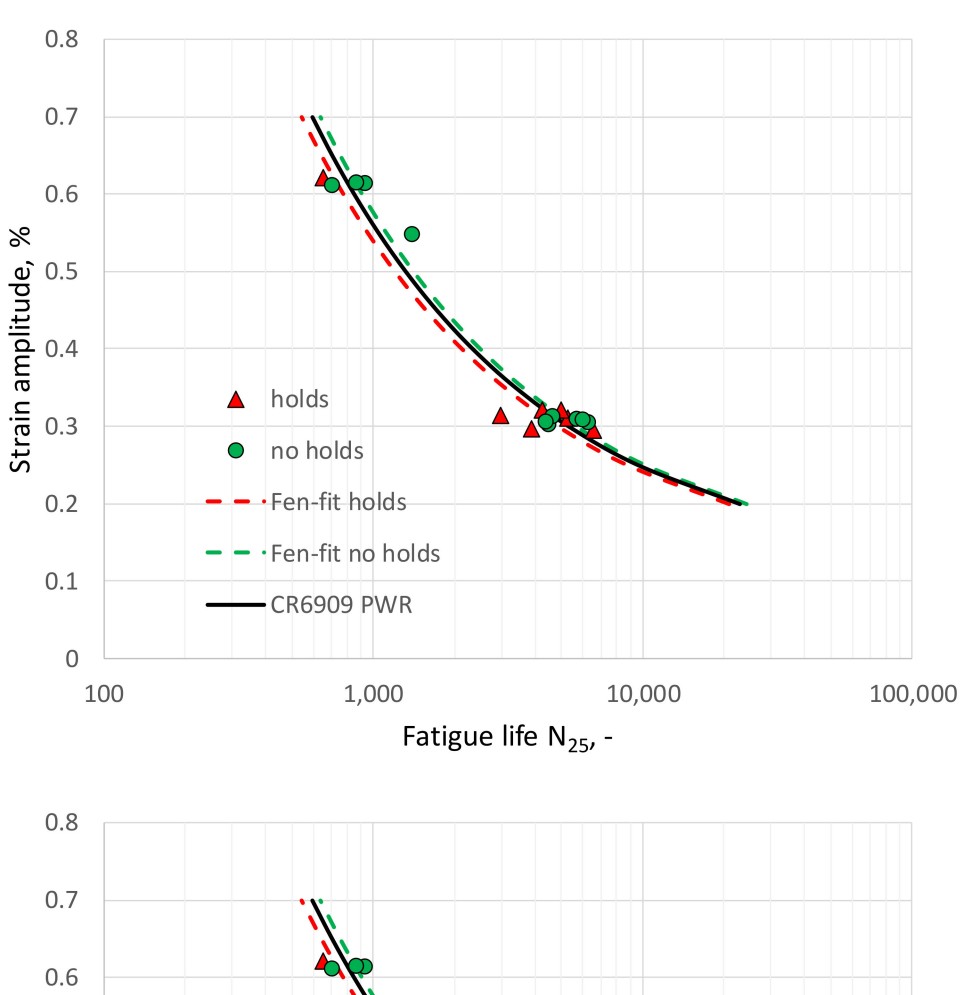

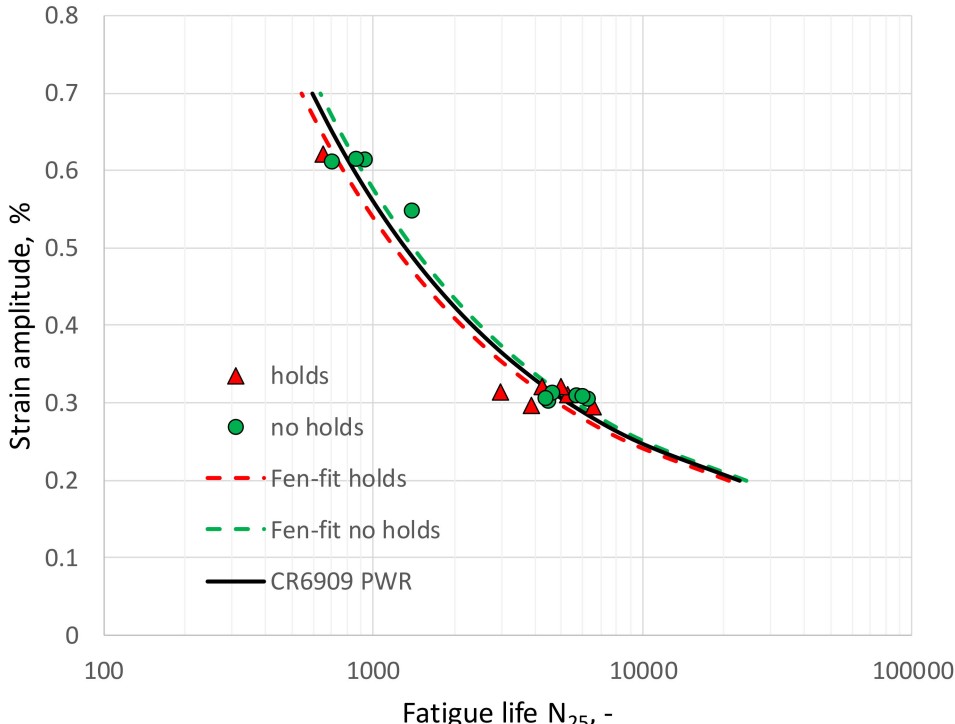

**Figure 8.** PWR fatigue life results (by holds) and NUREG/CR6909 mean PWR curve at 300 °C (Fen = 4.57).

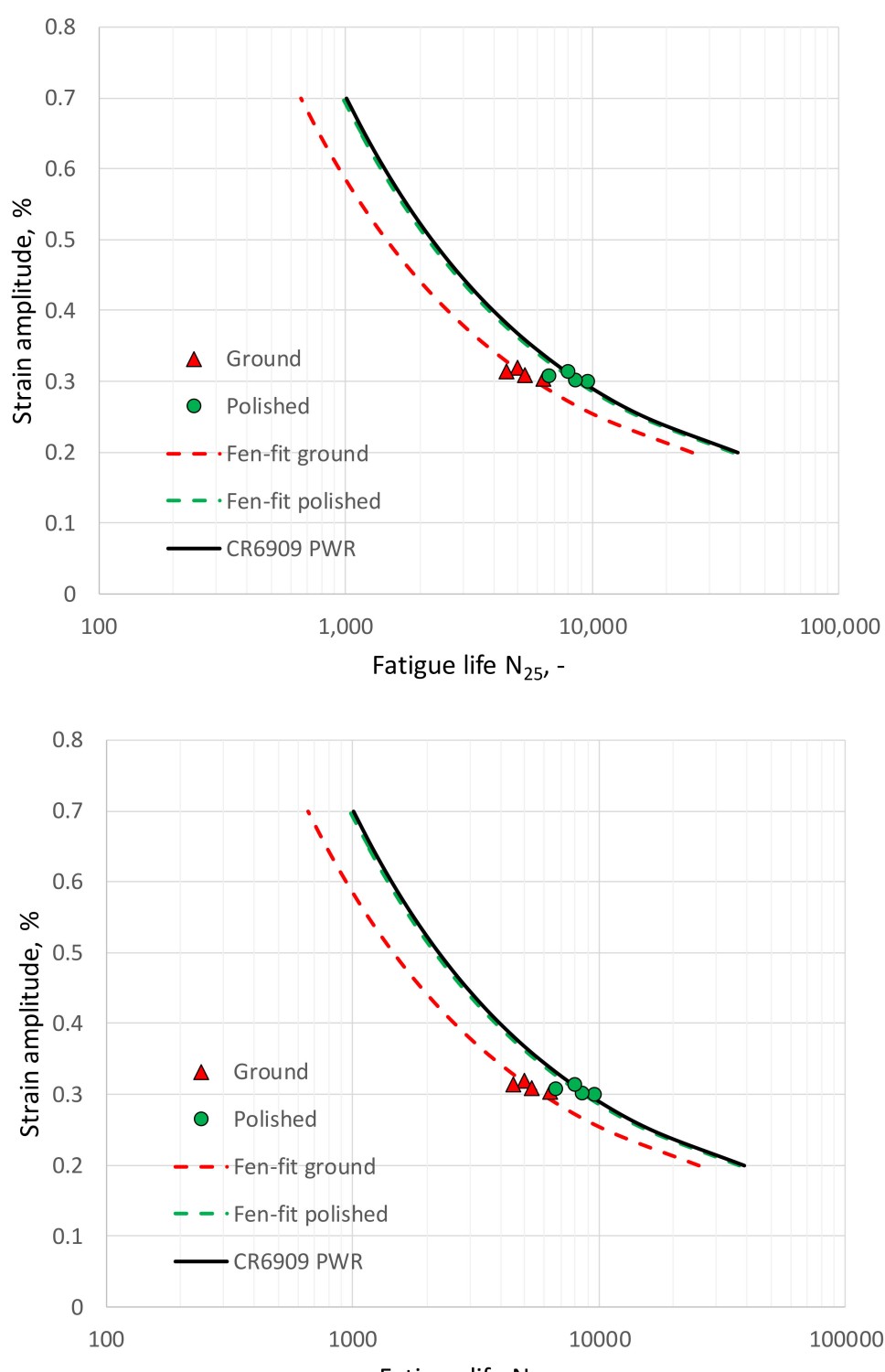

**Figure 9.** PWR fatigue life results (by surface finish) and NUREG/CR6909 mean PWR curve at 230 °C ($F_{en}$ = 2.69).

(3) At 300 °C, the fatigue life of the ground specimens is lower than the fatigue life predicted by NUREG/CR6909. Indeed, the 90% confidence interval for the "CR6909" factors on life are 1.10–1.32 for our results and 1.00–1.20 for the INCEFA-PLUS results. Both ranges lie below the provisions made by CR6909 for ground specimens, i.e., 1.5–3.5, when going from the mean curve to the design curve.

(4) At 230 °C, the fatigue life of the ground specimens is lower than the fatigue life predicted by NUREG/CR6909. Indeed, the 90% confidence interval for the "CR6909" factors on life are 1.42–1.68 for our results. The range lies at the low end of the provisions made by CR6909 for ground specimens, i.e., 1.5–3.5, when going from the mean curve to the design curve.

Table 6 shows experimental factors on life to be applied to NUREG/CR6909 life when going from the fatigue life of polished specimens to that of ground specimens. It must be noticed that the reduction in life to be applied is lower than what NUREG/CR6909 proposes. This conclusion is in agreement with that presented by INCEFA-PLUS partners in references [12,13] and references therein.

**Table 6.** Summary of surface finish factors.

| Factor Ground-to-Polished Fatigue Life | Our Fatigue Data | INCEFA-PLUS Fatigue Data |
| --- | --- | --- |
| Air 300 °C | – | 1.12 |
| PWR 300 °C | 1.37 | 1.21 |
| PWR 230 °C | 1.47 | – |

Note: NUREG/CR6909 range for surface finish 1.5–3.5.

These conclusions indicate that for the well-controlled material heat, surface finish and fatigue experiments of INCEFA-PLUS the design curve of NUREG/CR6909 is "conservative"; be it more so at 300 °C than at 230 °C and be it more so for a ground than for a polished surface finish. However, this would not necessarily mean that NUREG regulation could be relaxed. Indeed, one cannot guarantee the same to hold for other material heats and/or a less controlled surface finish.

*4.7. 'Gauge-Strain-Controlled' Fatigue Testing under PWR Conditions*

Since we installed both a shoulder and a gauge extensometer, we can examine the correlation between the amplitudes of both. In our environmental fatigue tests, the gauge extensometer was the controlling extensometer and the shoulder extensometer just yielded a response. The ratio between shoulder and gauge amplitude varies slightly from test to test. The average shoulder-to-gauge amplitude ratio at the start of a PWR fatigue test averages 1.51 (for 0.3% strain amplitude) and 1.46 (for 0.6% strain amplitude), varying between 1.39 and 1.65 (for 0.3% strain amplitude) and between 1.37 and 1.54 (for 0.6% strain amplitude) [21]. It was shown in [11] that shoulder-to-gauge amplitude ratio values at the start of the test approximately correspond to mid-life values for the INCEFA-PLUS material and temperatures.

Commonly, a shoulder extensometer is used to control an environmental fatigue test on solid specimens. Then the shoulder amplitude has to be estimated by a correlation to the gauge amplitude. The ratio shoulder-to-gauge amplitude does however depend on material, geometry and strain amplitude. Installing both extensometers simultaneously has shown that the ratio of the shoulder-to-gauge amplitude is relatively constant during an environmental fatigue tests; it only slightly differs at the beginning and towards the end of the test [11]. These measured shoulder-to-gauge amplitude ratios have been used to qualify experimental or improve FEM-calculated correlation methods as discussed in [21].

## 5. Conclusions

Environmental and air fatigue test have been performed on 304 austenitic stainless steel in the framework of the INCEFA-PLUS project.

The results indicate no influence of three long holds or a mean strain on fatigue life and an increased fatigue life for the polished specimen as compared to the rough specimens. This conclusion is reached by analyzing our environmental fatigue lives and comparing them to NUREG/CR6909 predictions.

The obtained fatigue life agrees well with the NUREG/CR6909 LWR mean curve; be it that the smooth specimen shows a slightly higher fatigue life and the ground specimens a lower fatigue life. However, the allowance made for surface finish by NUREG/CR6909, a factor 1.5–3.5, seems to be too conservative in view of our fatigue results and the analyzed INCEFA-PLUS fatigue results. However, this would not necessarily mean that NUREG regulation could be relaxed. Indeed, one cannot guarantee the same to hold for other material heats and/or a less controlled surface finish.

The use of a dual extensometer system (with control by the gauge extensometer) in all our PWR fatigue experiments allowed us to conclude that, for those strain-controlled fatigue testers having to rely on a shoulder extensometer (1), a shoulder-to-gauge calibration can be performed around mid-life (experimental option) or (2) a shoulder-to-gauge calculation can be performed at the start of the fatigue test where one does not have to include cyclic softening or hardening.

**Author Contributions:** Data curation, Formal analysis, Project administration, Supervision and Writing—original draft, M.V.; Funding acquisition, M.V., M.D.S. and C.M.; Writing—review & editing, M.D.S. and C.M. All authors have read and agreed to the published version of the manuscript.

**Funding:** The INCEFA-PLUS project has received funding from the Euratom Research and Training Programme 2014–2018 under Grant Agreement No. 662320. SCK CEN received additional financing from ENGIE under Contract Number CO-90-13-3212-00. Specimen preparation was cosponsored by EDF and Framatome.

**Acknowledgments:** The author gratefully acknowledge the technical staff at SCK CEN involved in equipment design and test execution, in particular Pierre Marmy and Luc Bens.

**Conflicts of Interest:** The authors declare no conflict of interest.

## Abbreviations

| | |
|---|---|
| AISI | American Iron and Steel Institute |
| AFNOR | Association Française de Normalisation |
| ASTM | American Society for Testing and Materials |
| BS | British Standard, publication series |
| EAF | Environmentally-Assisted Fatigue |
| EPRI | Electric Power Research Institute |
| INCEFA-PLUS | Increasing safety in NPPs by Covering gaps in Environmental Fatigue Assessment |
| ISO | International Organization for Standardization |
| LTO | Long Term Operation |
| $N_{25}$ | Fatigue life determined by a 25% tensile load drop |
| NUREG | United States Nuclear Regulatory Commission, publications series |
| PID | Proportional-Integral-Derivative |
| PWR | Pressurized Water Reactor |
| STP | Standard Temperature and Pressure |

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
