# Peer review of "Gauge-Strain-Controlled Air and PWR Fatigue Life Data for 304 Stainless Steel—Some Effects of Surface Finish and Hold Time"

_metals, doi:10.3390/met10091248_

Round 1

Reviewer 1 Report

This is a paper on the EAF test of 304L SS. It is a rather simple and clear test results and discussion. It seems OK, but the followings could be improved.

  • Authors used on heat (14XX) in air test and the other heat (SC) in PWR test. Could there be any heat-to-heat variability between the 2 heats of materials?
  • Throughout in the "Discussion", authors quantitatively compared the effect of surface finish and hold. Mostly, the ratio was less than 1.5, which, I think, is not significant considering the large scatter in EAF test data worldwide. Please justify such quantitative assessment.
  • In Table 1, the composition of test materials were provided. First, C content of both heats is close to 0.03% which is upper bound to be L-grade. The materials could be better described as regular-grade 304. Second, the S contents of 2 heats differ significantly (0.004 vs 0.021). Sulfur may have some effects on EAF life of SS. Could authors discuss such issue?

Author Response

Dear reviewer,

Thank you for your remarks.  Please find below my responses.

Remark 1:

Yes, of course there can be heat to heat variability between 14xx and SC-x.  The SC-x heat is a well controlled heat that is used through the wider INCEFA-PLUS project by all testing partners.  Since we did not participate extensively in the air testing part of the INCEFA-PLUS project, but we needed to perform air tests to receive matching funding, we had specimens made from an in-house, commercial heat.  We did measure the composition of the latter and, yes, it differs slightly from the common INCEFA-PLUS heat (please see also response to your third remark).  Our air fatigue tests do however fall relatively close to the CR6909 mean air curve.  A mean air curve which has been obtained for various stainless steel materials.  When going from the mean air curve to the design curve, see CR6909, heat-to-heat variability is part of the factor for data scatter and material variability, which has a factor of 2 in CR6909.

I have added the following sentence after the third paragraph in 2.1 material.

"Other elements are not considered in [20] for affecting either air or environmental fatigue life within the stainless steel material group.  Hence, we assume their influence on the air fatigue results here to be negligible."

Remark 2:

Please note the sentence [Line 230-234 These conclusions indicate that for the well-controlled material heat, surface finish and fatigue experiments of INCEFA-PLUS the design curve of NUREG/CR6909 is “conservative”; be it more so at 300 °C than at 230 °C and be it more so for a ground than for a polished surface finish.  However, this would not necessarily mean that NUREG regulation could be relaxed.  Indeed, one cannot guarantee the same to hold for other material heats and/or a less controlled surface finish.] in the manuscript, which already addressed the concern you raised.

Nevertheless, I have added some text related to statistics; i.e. which effects showed to be significant in the discussed data sets?

4.2, first bullet, end first sentence: "and this difference is found to be statistically significant."

4.2, second bullet, end first sentence: "but this difference is not found to be statistically significant."

4.2, end: "The statistical analysis is performed on a factor distribution, one for each test results.  The factor is the ratio of the test life, N25, to the CR6909 expected life,  (see equation 1).  It is assumed that the natural logarithm of the factor is normally distributed.  Further conclusions are that the polished, hold and non-hold data subsets are not statistically significant different from the CR6909 mean air curve but that the ground data subset is."

In section 4.6 we stated "Here we report on findings which, based on a simple t-test, yield significant effects."  Here, we have added an extra sentence "The statistical analysis is performed as describe at the end of section 4.2".

Finally, accepting your concern on scatter, we have included the following sentence of the manuscript again at the end of the third paragraph in the conclusions: "However, this would not necessarily mean that NUREG regulation could be relaxed.  Indeed, one cannot guarantee the same to hold for other material heats and/or a less controlled surface finish.".

Remark 3:

According to CR6909 the sulfur content of the material can affect EAF fatigue life in case of low alloy steel.  The effect is however mentioned in the environmental correction factor Fen and only for low alloy steel, not for stainless steel.  The air ASME and CR6909 curves do not consider the effect.

I have added the following sentence after the third paragraph in 2.1 material

"One element, sulfur, has been considered for affecting environmental fatigue life in [20] but only for carbon and low alloy steel and only through the environmental correction factor Fen. Hence, we assume its influence on the air fatigue results here to be negligible."

Regarding the carbon content, I have removed the L.  The reader can then decided, from the actual composition in the table, whether he wants to consider this material belonging to the low carbon grade or the regular grade.

I have added the specifications for 304 and 304L to the table and referred to them in the text.

Hoping we have addressed your remarks sufficiently, we remain,

yours sincerely,

Reviewer 2 Report

Reviewer remarks

Article:

PWR & AIR fatigue testing at SCK CEN in the framework of INCEFA-PLUS

The paper contains a presentation of fatigue test results of 304L steel in different conditions and under different strain control (with and without holds). The authors clearly present the experimental data and draw adequate conclusions. I find this work worth publishing after some small changes. My remarks can be found below.

Remarks:

Line 55 Although I understand that the specimen geometry can be found in different paper, this manuscript is a separate work and something as important as specimen geometry should be included within this paper. Also a whole subsection (2.2) being one sentence doesn’t feel right. Please expand this part of the paper.

There are to many abbreviations.  Please rewrite some of the sentences to substantially decrease the use of PWR and especially SCK CEN. At some point a reader might start to think he is reading a brochure/commercial not a scientific paper. This is also true for the paper title. Does SCK CEN stands for some kind of testing method? If so please explain this in the manuscript.

Figure 3 would be much clearer if the text next to each point was horizontal.

The reference list is quite short. Maybe this could be extended by adding something like ‘state of the art’ section in the introduction?

Regards,

Author Response

Dear reviewer,

Thank you for your remarks.  Please find below my responses.

Remark 1: specimen geometry

The subsection 2.2 has been expanded and includes a figure of the specimen geometry.

Remark 2: abbreviations

References to SCK CEN, the first author's institute, have been removed as far as possible.  We have changed SCK CEN to we/our or removed it altogether.  We were majorly concerned with distinguishing between our fatigue results and those by all partners withing the iNCEFA-PLUS project for, to the latter, we do not hold copyright.

The removal of PWR is a bit more difficult.  We need to ensure that we distinguish adequately between fatigue results in air and fatigue results in PWR.  Upon its first occurance, PWR is spelled out as pressurized water reactor and later the abbreviation is used.

Remark 3:

The labels have been removed in the figure 3 (original).  Indeed, labels seems to overlap, whatever direction of the label text.  Actual values are listed in the results table, which allows a reader to identify the points in the figure anyway.

Remark 4:

The following sentence has been added to the introduction section, at the end of the first paragraph.

"Current regulation is based on a state-of-the-art review of light water reactor fatigue data and effects [20]"

The following sentence has been added to the introduction section, at the end of the second paragraph.

"... and mentioned as gaps to be addressed in [2-4]"

Hoping we have addressed your remarks sufficiently, we remain,

yours sincerely,

Reviewer 3 Report

It is high quality paper about PWR & AIR fatigue testing at SCK CEN. No corrections for the improvement are needed.

Author Response

Dear reviewer,

Thanking you for your remark, we remain,

yours sincerely,

Reviewer 4 Report

These are comments regarding the manuscript “PWR & AIR fatigue testing at SCK CEN in the framework of INCEFA-PLUS” submitted to be evaluated for publishing on journal “Metals”.

The manuscript reports fatigue experimental results for the stainless steel 304L with reference to their usage in nuclear power plants. Different aspects were considered, such as: roughness, environmental effects (reactor’s water), holding times, … Results were compared with database data and a good discussion was provided. The overall merit is good and it could be accepted for publishing.

Besides these positive considerations, some negative ones are here highlighted. First of all, it is opinion of this reviewer that this manuscript is closer to a “Short communication” rather than to a full “Article”. If authors are fine with this modification, the manuscript requires just some minor improvements as those suggested here:

  • acronyms should be avoided in titles were, generally, only well-known acronyms are used. On the contrary, this title could be incomprehensible to mostly all the potential readers. Please consider to modify it according to this.
  • introduction should be improved to give the reader a wider perspective. See for example the following articles referring to applications in this research field:
  • Fellinger, et al. Overview of fatigue life assessment of baffles in Wendelstein 7-X, Fusion Eng. Des., 136 (2018), pp. 292-297.
  • Citarella, V. Giannella, M. A. Lepore, J. Fellinger, FEM-DBEM approach to analyse crack scenarios in a baffle cooling pipe undergoing heat flux from the plasma, AIMS Materials Science, 2017, 4(2): 391-412.
  • Giannella, V.; Citarella, R.; Fellinger, J.; Esposito, R. LCF assessment on heat shield components of nuclear fusion experiment “Wendelstein 7-X” by critical plane criteria. Procedia Struct. Integr. 2018, 8, 318–331
  • Lines 55-56: that is not true. Please add a figure showing the specimen geometry, information on surface finish, etc. The continuous recalling of references should be avoided to improve readability of this manuscript.
  • Line 64: “roughness with an Ra of 0.60 μm”. Please add a comment.
  • Line 82: why did you consider holding times? Is it typical for these applications? What does it would represent? Please add comments.
  • Lines 99-100: as for Line 55, please add figures of extensometers to improve readability.
  • Line 103: test matrix does not represent a result. It should be inserted somewhere above within the manuscript.
  • Line 111: please add reference for equations.
  • Line 133: please consider to rephrase this sentence. The authors should also clarify why they were expecting an opposite behaviour. Do you have any references about that?

If authors are not fine with the modification from Article to Short communication, the manuscript has to be significantly improved in several aspects. Besides those proposed above, please consider that a more detailed introduction should be provided, with several references not only coming from INCEFA-PLUS framework. Comparisons with further literature data, potential applications, etc. should be also provided.

Author Response

Dear reviewer,

Thank you for your remarks.  Please find below my responses.

Main remark:

We accept the publication as a "short communication" rather than a full "article".  There is a shortage of open literature, numeric data in respect of EAF related to USNRC/CR6909.  Hence, the open publication of our data, gives a valuable contribution to open access data that can be compared to USNRC/CR6909.  USNRC/CR6909 together with the EPRI EAF gap analysis and roadmap reports embody the state-of-the-art in terms of EAF for light water reactors.

Bulleted remarks:

Bullet 1

There are two acronyms in the title, PWR and SCK CEN.

References to SCK CEN, the first author's institute, have been removed as far as possible.  We have changed SCK CEN to we/our or removed it altogether.  We were majorly concerned with distinguishing between our fatigue results and those by all partners withing the INCEFA-PLUS project for, to the latter, we do not hold copyright.

The removal of PWR is a bit more difficult.  We could replace PWR by environmental, if really needed, but this would be less specific.

We have changed the title to "Gauge strain controlled air and PWR fatigue life data for 304 stainless steel – some effects of surface finish and hold time"

Bullet 2

In the introduction it has been outlined that SCK CEN performs this air and PWR testing in the framework of an international, collaborative project on environmetally-assisted fatigue, INCEFA-PLUS.  Hitherto reference has been made to

  • INCEFA-PLUS publications
  • The EPRI EAF gap analysis and roadmap reports
  • The NUREG/CR6909 report

These references set the scene for the work performed.

Bullet 3-5

The papers referred to by the reviewer (bullets 3-5) concern fatigue evaluations on fusion components.  They address evaluations with respect to fatigue crack growth (majorly), fatigue initiation life or fatigue (thermal) loading characterization by FEM and component testing.  They do not address experimental work to address fatigue initiation life for use in regulatory approaches.  As such they are a bit out of the scope of the submitted article/communication.

Bullet 6

I am sorry, but lines 55-56 (original) are correct.  Reference [10] does contain the specimens geometry, see p227.  Nevertheless, subsection 2.2 has been expanded and includes a figure of the specimen geometry.

Bullet 7

Line 64 (original) has been modified to include after Ra "(arithmetic mean value)" and after measured "by a Handysurf E-35B instrument".

Bullet 8

Line 82 (original).  Hold times were included in the international, collaborative INCEFA-PLUS project because the EPRI gap analysis report and roadmap on environmentally-assisted fatigue listed this as an issue to address by getting more fatigue life data.  The holds represent the holding of primary circuitry at nominal conditions for extensive periods during actual operation.

The following sentence has been added at the end of section 2.5 "Holds were included into the inter-laboratory, INCEFA-PLUS test matrix to represent fatigue that is interrupted by holding for extensive periods of time at nominal plant conditions [6-9], an area for which there is a lack of data [2,4].".

Bullet 9

Line 99-100 (original).  I am sorry, but this part was removed after mdpi ran a similarity check with reference [10] and concluded that sections that were discussed in [10] should be referred to in this article/communication.  Hence, I have not performed the inclusion as you suggest but will raise it with mdpi in the accompanying letter to the editor.  Of course, you are also welcome to raise the issue with them directly.  Note that the location of the gauge and shoulder extensometer are however shown in the figure related to geometry (new figure 1), now included.

Bullet 10

Line 103 (original).  The text "The test matrix for SCK CEN's air and environmental fatigue testing is given in Table 5, together with the test results in terms of N25 fatigue life. "  has been changed to "Our air and environmental fatigue test results are listed in Table 5, in terms of N25 fatigue life. ".  The fatigue tests performed at SCK CEN were part of a test matrix set up by the international, collaborative INCEFA-PLUS project.

Bullet 11

Line 111.  The reference has been given in the text referring to equations (1) and (2), i.e. [20].  The sentence has been slightly improved from

"...[20].  There, mean fatigue life in air and up to 400 °C for austenitic stainless steel is given by equation (1).  Mean fatigue life in a PWR environment is obtained by applying an environmental correction factor Fen, as illustrated in equation (2). " to

"...[20].  There, the mean fatigue life in air and up to 400 °C for austenitic stainless steel is given by equation (1) and the mean fatigue life in a PWR environment is obtained by applying an environmental correction factor Fen, as illustrated in equation (2). ".

Bullet 12

Line 133.  The sentence has been rewritten as follows:

"The obtained N25 fatigue lives lie close to the NUREG/CR6909 mean air curve (Figure 3).  This was not expected since the specimens had a machined surface finish, whilst the mean air curve is based on fatigue tests with polished samples.  Polished samples have a lower surface roughness than machined samples (here Ra = 0.044 and 0.6 µm, respectively).  Given that the machining marks were clearly visible and radial on the machined specimens and that the ground INCEFA-PLUS specimens show a reduced fatigue life (having a surface roughness between 1.1 and 5.1 µm Ra), one could have reasoned machined life to be closer to ground life than polished life on the basis of surface roughness."

Hoping we have addressed your remarks sufficiently, we remain,

yours sincerely,

Round 2

Reviewer 1 Report

The revised version seems OK. Thank you.

Reviewer 4 Report

Authors improved the manuscript according to the suggestions of reviewers. It can be accepted for publishing now. Please consider that, according to the suggestion of this reviewer, then accepted by authors, line 1 should be modified from "Article" to "Short communication".

Some optional suggestions for minor refinements are in the followings:

  • articles are generally written impersonally. "we/our performed..." are generally avoided, preferring "it has been performed...". As instance, in table 6, rather than "Our data" -> "Current work".
  • figure 1 should be enlarged so as to have higher redability; it is hard to read some figures at 100% zoom.